# MIMIC: MASKED IMAGE MODELING WITH IMAGE CORRESPONDENCES

## ABSTRACT

Dense pixel-specific representation learning at scale has been bottlenecked due to the unavailability of large-scale multi-view datasets. Current methods for building effective pretraining datasets heavily rely on annotated 3D meshes, point clouds, and camera parameters from simulated environments, preventing them from building datasets from real-world data sources where such metadata is lacking. We propose a pretraining dataset-curation approach that does not require any additional annotations. Our method allows us to generate multi-view datasets from both real-world videos and simulated environments at scale. Specifically, we experiment with two scales: MIMIC-1M with 1.3M and MIMIC-3M with 3.1M multi-view image pairs. We train multiple models with different masked image modeling objectives to showcase the following findings: Representations trained on our automatically generated MIMIC-3M outperform those learned from expensive crowdsourced datasets (ImageNet-1K) and those learned from synthetic environments (MULTIVIEW-HABITAT) on two dense geometric tasks: depth estimation on NYUv2 (↑1.7%), and surface normals estimation on Taskonomy (↓2.05%). For dense tasks which also require object understanding, we outperform MULTIVIEW-HABITAT, on semantic segmentation on ADE20K (↑3.89%), pose estimation on MSCOCO (↑9.4%), and reduce the gap with models pre-trained on the object-centric expensive ImageNet-1K. We outperform even when the representations are frozen, and when downstream training data is limited to few-shot. Larger dataset (MIMIC-3M) significantly improves performance, which is promising since our curation method can arbitrarily scale to produce even larger datasets.

## 1 INTRODUCTION

Today, dense vision tasks—depth prediction, surface normal estimation, semantic segmentation, and pose estimation— often rely on pretrained representations (He et al., 2022; Bachmann et al., 2022). Naturally, self-supervised learning lends itself as a potential solution. Despite the impressive performance on object recognition and other high-level tasks, self-supervised representations for dense prediction tasks have not yet fully delivered (Weinzaepfel et al., 2022). The representations trained on object-centric datasets such as ImageNet-1K (Deng et al., 2009) do not transfer well to dense prediction datasets such as NYUv2 (Silberman et al., 2012), and KITTI (Geiger et al., 2012), Cityscapes (Cordts et al., 2016), which contain indoor and outdoor scenes. Moreover, the joint-embedding-based objectives (SimCLR (Chen et al., 2020), MoCo (He et al., 2020), DINO (Caron et al., 2021)) that are often used on object-centric datasets utilize augmentations that do not preserve geometric pixel-wise information. In response, the general purpose representation learning method—masked image modeling and specifically masked autoencoders (MAE)—has become a popular default mechanism for such tasks (He et al., 2022; Bachmann et al., 2022; Weinzaepfel et al., 2022). Unfortunately, recent findings suggest that the representations learned by MAE are devoid of sufficient local information for tasks like depth estimation (Weinzaepfel et al., 2022).

Based on these observations, we ask the following question: *What data do we need to learn useful representations for dense vision tasks*? We find a potential answer in cognitive science: 3D understanding of the physical world is one of the first visual skills emergent in infants; it plays a critical role in the development of other skills, like depth estimation, understanding surfaces, occlusions, etc (Held & Hein, 1963). Scientists hypothesize that 3D understanding emerges from infants learning the relationship between changes in visual stimuli in response to their self-motion (Jayaraman

Figure 1: We introduce a data-curation method that generates multi-view image datasets for self-supervised learning. Our method identifies potential data sources, including videos of indoor scenes, people, and objects, 3D indoor environments, outdoor street views, and stereo pairs to mine potential multiview images. Next, we use classical computer vision methods such as SIFT keypoint detection and homography transformation to locate corresponding patches. Finally, we filter pairs based on a threshold for significant overlap, ensuring a substantial percentage of pixels match between a pair.

& Grauman, 2015), i.e. 3D awareness emerges by learning correspondences between appearances as the infant's vantage point changes (Rader et al., 1980).

Very recently, a machine learning paper proposed a variant of masked image modeling, named **cro**ss-view **co**mpletion (CroCo), which uses an objective that operationalizes learning representations in response to changes in self-motion (Weinzaepfel et al., 2022). Given a pair of multi-view images, CroCo reconstructs a masked view using the second view as support. Unfortunately, CroCo is a data-hungry objective. Its synthetic MULTIVIEW-HABITAT dataset of 1.8M multi-view images was curated using a method that requires ground truth 3D meshes to be annotated. Although CroCo shows promise, the lack of datasets with 3D annotations is a severe limitation, preventing its objective from scaling. If one could mine large-scale multi-view datasets, perhaps dense vision tasks could enjoy the success that the field of natural language processing has welcomed due to the availability of large-scale pretraining text (Brown et al., 2020).

In this work, we contribute MIMIC: a data-curation method for developing multi-view datasets that scale. Our method does not require any 3D meshes and can generate multi-view datasets from unannotated videos and 3D simulated environments. We leverage classical computer vision techniques, such as SIFT(Scale Invariant Feature Transform) keypoint detection (Lowe, 2004), RANSAC (Fischler & Bolles, 1981), homography estimation (Hartley & Zisserman, 2003), etc. to extract correspondences between frames in open-sourced unannotated videos (see Figure 1). In other words, MIMIC produces a pretraining dataset for **m**asked **i**mage **m**odeling using **i**mage **c**orrespondences.

We experiment with two scales: MIMIC-1M and MIMIC-3M, and show that they effectively train useful self-supervised (MAE and CroCo) representations when compared to MULTIVIEW-HABITAT. Our experiments show the following: Most importantly, representations learned from MIMIC-3M, our automatically generated dataset, outperform those trained using ImageNet-1K Deng et al. (2009), an expensive human-labeled dataset on dense geometric tasks: depth estimation (NYUv2 (Nathan Silberman & Fergus, 2012)) and surface normals (Taskonomy (Zamir et al., 2018)); Second, MIMIC also trains better representations than MULTIVIEW-HABITAT Weinzaepfel et al. (2022), a baseline automatically generated dataset, on both dense geometric tasks, such as depth estimation (NYUv2) and surface normal prediction (Taskonomy), as well as on dense object-related tasks, such as semantic segmentation (ADE20K (Zhou et al., 2019)) and pose estimation (MSCOCO (Lin et al., 2014)). Third, larger pretraining dataset (MIMIC-3M > MIMIC-1M) significantly improves performance, which is promising since our curation method can arbitrarily scale to produce even larger datasets. Finally, our representations demonstrate better few-shot performance on depth estimation (NYUv2) and semantic segmentation (ADE20K).

## 2 RELATED WORK

In this section, we discuss masked image modeling - a promising paradigm for self-supervised dense representation learning at scale and data curation methods for large-scale visual learning.

**Masked image modeling.** Amongst masked image modeling, BEiT (Bao et al., 2021) proposes the pre-training task of recovering the visual tokens from a corrupted image, MAE (He et al., 2022) learns by masking patches of an image and inpainting the masked patches; MultiMAE extends MAE to a multi-task formulation (Bachmann et al., 2022). Their approach uses pseudo-labels–

hence, MultiMAE is not fully self-supervised. CroCo (Weinzaepfel et al., 2022) uses cross-view completion and ingests multi-view images. Their data curation method, though, uses 3D metadata and meshes of synthetic 3D environments; their dataset is also not publicly available. By contrast, MIMIC neither needs any pseudo labels extracted using supervised methods nor it needs any 3D meshes, point clouds, or camera parameters for dataset curation.

**Data curation for large scale visual learning.** Large-scale image datasets have incredibly accelerated progress in visual learning. ImageNet-1K, with 1.2M images annotated by crowdsourcing led to several breakthroughs and is still a standard dataset used for pretraining vision models. It was manually designed to cover a diverse taxonomy of object categories with sufficient representation of instances per category. Unfortunately, this approach is extremely costly, not scalable, and serves as an upper bound for what is possible with manual curation instead of our automatic curation.

Moreover, the efforts so far have been focused on high-level semantic tasks like classification, and large-scale pretraining datasets for dense prediction tasks such as MULTIVIEW-HABITAT with synthetic image pairs mined using Habitat simulator Savva et al. (2019) are not available publicly. MULTIVIEW-HABITAT uses annotations such as camera parameters and meshes to sample image pairs with a co-visibility threshold of 0.5. The use of such metadata for mining image pairs is a limiting factor as (1) it requires expensive sensors to obtain these annotations on real-world datasets (2) it cannot be scaled up to mine web-scale data sources where this information is not readily available.

To address these challenges we propose a methodology for curating multi-view datasets using videos and 3D environments. We demonstrate that it is possible to use our data collection strategy and outperform on multiple dense vision tasks without making use of any explicit annotations.

## 3 MIMIC: CURATING MULTI-VIEW IMAGE DATASET FOR DENSE VISION TASKS

While CroCo recently utilized MULTIVIEW-HABITAT, a multi-view dataset, their dataset creation process requires the availability of 3D mesh, point cloud, or camera pose information for each scene. This dependency imposes limitations on the range of data sources that can be used for crafting a multi-view dataset. Unfortunately, there is currently no large-scale publicly available dataset to address this void. To bridge this gap, we introduce MIMIC.

MIMIC can generate multi-view datasets from unannotated videos and 3D simulated environments. Any data source that contains multi-view information with static objects or at least with minimal object movement is a suitable data source. MIMIC works by cleverly combining traditional computer vision methods (Figure 1). The only mechanism our curation process requires is a sampling mechanism $(I_1, I_2) \sim g(S)$, where $S$ is some data source from which $g(\cdot)$ samples two images $I_1$ and $I_2$. For example, $S$ can be a video from which $g(\cdot)$ samples two image frames. Or $S$ can be a synthetic 3D environment from which $g(\cdot)$ navigates to random spatial locations and samples two random image renderings of the same scene.

**Identifying data sources.** We generate our MIMIC dataset from both real as well as synthetic data sources. We use DeMoN (Ummenhofer et al., 2017), ScanNet (Dai et al., 2017), ArkitScenes (Baruch et al., 2021), Objectron (Ahmadyan et al., 2021), CO3D (Reizenstein et al., 2021), Mannequin (Li et al., 2019), and 3DStreetView (Zamir et al., 2016) as real data sources. DeMoN is a dataset containing stereo image pairs. ScanNet and ArkitScenes contain videos from indoor environments. Objectron and CO3D are collections of videos containing objects. Mannequin provides a video dataset featuring individuals engaged in the mannequin challenge. 3DStreetView offers a collection of street images from multiple urban areas.

We also source data from 3D indoor scenes in HM3D (Ramakrishnan et al., 2021a), Gibson (Xia et al., 2018), and Matterport (Chang et al., 2017) datasets, using the Habitat simulator (Savva et al., 2019). We initialize an agent randomly in the 3D environment and design $g(\cdot)$ to move the agent in random steps and directions. For each scene, the agent moves to numerous locations and captures various views. All our data sources with their distributions are visualized in Figure 2.

**Mining potential pairs.** The primary characteristic of the image pairs in our dataset resides in their ability to capture the same scene or object from varying viewpoints while exhibiting a substantial degree of overlap. The dataset is designed to strike a balance: the overlap is not excessively large to

the point of containing identical images, rendering the pre-training task trivial; nor is it excessively small, resulting in disjoint image pairs that offer limited utility, making the task only self-completion. Particularly, we discard the image pairs with a visual overlap of less than 50% and more than 70%. We base this design decision on empirical ablations performed in CroCo. Their experiments suggest that cross-view completion offers no advantage if the visual overlap is outside of this range.

In each video or scene, many image pairs can be generated. However, we focus on selecting a limited number of pairs that are more likely to meet our desired condition of having sufficient overlap. Nonetheless, not all of these candidate pairs may ultimately be chosen. For instance, when dealing with video data, a practical strategy involves creating a list of frames at regular time intervals, which depends on the video's speed. By selecting consecutive frames from this list, potential pairs are generated. Conversely, collecting potential pairs in 3D scenes such as HM3D (Ramakrishnan et al., 2021a) or Gibson (Xia et al., 2018) presents greater challenges. Therefore, inspired by CroCo, we employ the habitat simulator (Savva et al., 2019) to capture comprehensive environment views. The agent undergoes random rotations and movements, exploring the scene from various perspectives. By capturing images during these random walks, we generate potential pairs for further analysis. The selection process involves filtering based on a specified overlap range (50% to 70%) and ensuring the inclusion of pairs with diverse viewpoints. However, our approach does not rely on additional annotations and solely utilizes the available images.

**Matching and measuring overlap.**
Given a potential image pair capturing a scene, we employ the robust, and widely used SIFT features to localize key points in both images.

After obtaining the key points and descriptors, we apply a brute-force matching technique to establish correspondences between the key points in the first image and those in the second image. More efficient methods, such as FLANN matcher (Muja & Lowe, 2009), may offer ($\approx 1.24\times$) speedups. However, our initial exploration shows that brute-force matching yields better matches; also, extracting pairs is a one-time process. We further utilize these matches to estimate the homography matrix, us-

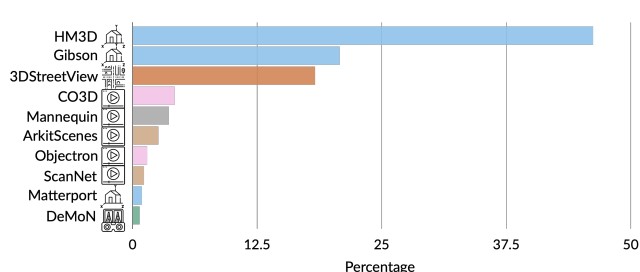

Figure 2: Distribution of Data Sources (%). Real data sources, including DeMoN, ScanNet, ArkitScenes, Objectron, CO3D, Mannequin, and 3DStreetView, contribute to 32% of MIMIC. The remaining portion consists of synthetic sources, namely HM3D, Gibson, and Matterport.

ing the RANSAC (Random Sample Consensus) algorithm to eliminate outliers. Note that the homography transformation holds true in three scenarios–(1) when capturing planar surfaces, (2) when capturing a distant scene, and (3) when a camera undergoes a pure rotation. In real-world videos, these assumptions may not always hold true. Regardless, homography serves as an approximation to the transformation. We further use this approximated matrix to filter out unwanted image pairs with no visual overlap.

We then partition each image into non-overlapping patches of size $N \times N$, where $N$ is the patch size of the image encoder. We use $N = 16$ for our experiments. For each patch in the first image, we search for the corresponding patch in the second image with the highest overlap. We randomly sample points within the first image and match them with their correspondences in the second image. Next, we map each patch in the first image to the patch with the highest number of matched correspondences in the second. Lastly, we measure visual overlap by calculating the total number of matched patches divided by all patches. Refer to the Appendix for more details.

**Filtering out degenerate matches.** In our approach, the selection of image pairs is guided by the objective of capturing shared 3D information while mitigating redundancy. Hence, the desired pairs consist of images that depict the same objects or scenes from different perspectives. This characteristic enables the learning model to acquire valuable insights about the underlying 3D structure. However, it is crucial to avoid including pairs where one image is a zoomed-in version of the other, as such pairs provide limited additional information.

To address this concern, we modify the overlap metric used in the pair selection process. Specifically, we incorporate a criterion that prevents the inclusion of patches from the first image that have exact correspondences in the second image. Therefore, in the counting, we consider all patches that have the same corresponding patch in the second image as a single entity.

**Overall statistics.** To understand the effect of data size we experiment with two scales. MIMIC-1M, comprises a total of $1,316,199$ image pairs, each capturing different scenes or objects from varying viewpoints. Among these pairs, $761,751$ are sourced from HM3D, $305,197$ from Gibson, $29,658$ from Matterport, $114,729$ from Mannequin, $22,184$ from DeMoN, $36,433$ from ScanNet, and $46,250$ from Objectron. We further expand the dataset to create a larger version, MIMIC-3M, to contain a total of $3,163,333$ image pairs. This expansion involves augmenting the HM3D dataset with an additional $699,322$ pairs, the Gibson dataset with $351,828$ pairs, and the inclusion of new datasets such as ArkitScenes with $81,189$ pairs, CO3D with $133,482$ pairs, and 3DStreetViews with $579,310$ pairs. By incorporating these new datasets, we further enrich the diversity and quantity of image pairs available in our dataset.

# 4 TRAINING WITH MIMIC

To measure the effectiveness of MIMIC, we train two models with masked image modeling objectives and evaluate the utility of the learned representations on downstream dense prediction tasks. We compare against existing pretraining dataset alternatives.

## 4.1 PRETRAINING

We use MAE (He et al., 2022) and CroCo (Weinzaepfel et al., 2022) for pretraining. We follow the protocol from CroCo and use a ViT-B/16(Dosovitskiy et al., 2020) as a backbone for all our experiments with input images sizes of $224 \times 224$. We train our models on $8$ RTX A6000 GPUs for $200$ epochs with a warmup of $20$ epochs with a base learning rate of $1.5 \times 10^{-4}$, an AdamW (Loshchilov & Hutter, 2017) optimizer with a cosine learning rate schedule, a weight decay of $0.05$, and an effective batch size of $4096$. Lastly, we evaluate these pretrained representations on a series of downstream dense prediction tasks.

**MAE pretraining.** To understand the importance of including correspondences in the pretraining objective, we train MAE, which does not encode multi-view correspondences and treats each image in our image pairs independently. MAE masks out a large portion ($75\%$) of the input patches of an image and uses an asymmetric encoder-decoder architecture to reconstruct the masked-out pixels. Specifically, it uses a ViT-based encoder to extract the latent representations of the masked view. Then it pads the output with the masked tokens and feeds it to a lightweight decoder. The decoder's output reconstruction is optimized with an L2 loss. The reconstruction pixel targets are normalized by computing the mean and standard deviation of the image patches.

**CroCo pretraining.** Unlike MAE, CroCo aims to encode relationships between the two views of the same scene from different viewpoints and learns to reason about the illumination and viewpoint changes. CroCo reconstructs a masked image input similar to MAE but supports the reconstruction process through an unmasked second reference view. CroCo masks $90\%$ of the first image. CroCo uses the same ViT encoder as MAE, with shared weights to encode both views. The decoding cross-attends over the second view while reconstructing the first masked view.

## 4.2 BASELINE DATASETS.

We compare MIMIC with: ImageNet-1K (Deng et al., 2009) and MULTIVIEW-HABITAT (Weinzaepfel et al., 2022).

**ImageNet-1K** is a widely used large-scale dataset with $1.2$M training images. It was manually designed to cover a diverse taxonomy of a thousand object categories. The images were chosen to have sufficient instances per category. Therefore, ImageNet-1K serves almost as an upper bound for what is possible with immense human data-curation effort.

**MULTIVIEW-HABITAT** comprises of synthetic renderings of indoor scenes collected using the 3D meshes available in the Habitat simulator (Savva et al., 2019). It is derived from HM3D (Ramakrish-

Table 1: For **dense geometric tasks** including depth estimation and surface normals estimation, CroCo pretrained with MIMIC-3M outperforms MAE and DINO on ImageNet-1K as well as MULTIVIEW-HABITAT. We report the results from the CroCo paper (marked with *) as well as those with our task-specific fine-tuning setup adopted from MultiMAE.

| Model | Frozen | Dataset | NYUv2 depth est. | Taskonomy surface normal est. |
|---|---|---|---|---|
| | | | $\delta 1$ ($\uparrow$) | L1 ($\downarrow$) |
| DINO | ✗ | ImageNet-1K | 81.45 | 65.64 |
| MAE | ✗ | ImageNet-1K | 85.1 | 59.20 |
| MAE | ✓ | MV-HABITAT | - | - |
| MAE | ✓ | MIMIC-3M | 80.65 | 68.97 |
| MAE | ✗ | MV-HABITAT | 79.00 | 59.76 |
| MAE | ✗ | MIMIC-3M | **85.32** | **58.72** |
| CroCo | ✓ | MV-HABITAT | 85.20* (84.66) | 64.58 |
| CroCo | ✓ | MIMIC-3M | **85.81** | **61.7** |
| CroCo | ✗ | MV-HABITAT | 85.60* (90.19) | 54.13 |
| CroCo | ✗ | MIMIC-3M | **91.79** | **53.02** |
| | | | +1.6 | -1.11 |

nan et al., 2021b), ScanNet (Dai et al., 2017), Replica (Straub et al., 2019) and ReplicaCAD (Szot et al., 2021). This dataset is not available publicly. So, we compare against the released models trained on it. MULTIVIEW-HABITAT serves as our main baseline dataset since it is the only large-scale multi-view dataset that has been used for training use representations for dense vision tasks.

## 4.3 DOWNSTREAM TASKS, DATASETS, EVALUATION PROTOCOLS

We evaluate our models on two dense geometric tasks: depth estimation and surface normal estimation. We also evaluate on two dense object-related tasks: semantic segmentation, and pose estimation. Finally, we report object classification numbers for completion. We provide below the details of the datasets, metrics, and protocols used for fine-tuning and evaluations.

**Depth Estimation** involves estimating the depth of each pixel of an input image from the camera. For evaluation, we use the NYUv2 (Nathan Silberman & Fergus, 2012), a dataset of RGB images and their corresponding ground truth depth maps. It consists of 795 training and 654 test images of indoor scenes. We report the $\delta 1$ metric on the test images - which computes the percent of the pixels with error $max(\frac{y_{p_i}}{y_{g_i}}, \frac{y_{g_i}}{y_{p_i}})$ less than $1.25$, where $y_{p_i}$ is the depth prediction and $y_{g_i}$ is the ground truth of the $i$th pixel of an image. We use DPT (Ranftl et al., 2021) head as in MultiMAE for finetuning.

**Surface Normals Estimation** is a regression task that aims to estimate the orientation of a 3D surface. We use a subset of Taskonomy (Zamir et al., 2018) with 800 training images, 200 validation images, and $54,514$ test images. We use the L1 loss value on the test set as a metric for evaluations.

**Semantic Segmentation** entails assigning a class to each pixel of an image based on its semantic category. We use ADE20K (Zhou et al., 2019), which consists of $20,210$ training images and $150$ semantic categories. We report the mIOU which quantifies the percentage overlap between the predictions and the ground truth annotations. For finetuning, we use a segmentation head based on ConvNext (Liu et al., 2022) adapter.

**Classification** is a high-level semantic task that involves assigning a category to an image based on its content. We use ImageNet-1K(Deng et al., 2009) which contains $1.28M$ training images and 50K validation images. This task allows us to measure how large the gap is when models are pretrained for dense tasks in mind. We follow the linear probing protocol from MAE and report accuracy.

**Pose Estimation** involves detecting keypoints and their connections in an image. We use the MSCOCO (Lin et al., 2014) dataset for finetuning and report Average Precision and Average Recall on the validation set. Specifically, we adopt ViTPose-B (Xu et al., 2022) for finetuning.

# 5 EXPERIMENTS

We evaluate our pre-trained models on two dense geometric vision tasks – depth estimation and surface normal prediction. MIMIC-3M's dense representations outperform both tasks (§ 5.1). Next, we finetune our encoders for pixel-level tasks that also require object understanding – semantic segmentation, and pose estimation and high-level semantic tasks – image classification. For these three tasks, our experiments demonstrate that models trained using our automatically generated data close the gap with models trained on ImageNet-1K (§ 5.2). We further experiment with the data size used for pretraining and showcase that more data leads to improvements on depth estimation and semantic segmentation tasks (§ 5.3). Unlike CroCo trained on MULTIVIEW-HABITAT, our pre-trained models do not saturate or degrade over time on depth estimation and semantic segmentation (§ 5.4). Our performance benefits also hold as we vary the number of fine-tuning data points available for both depth estimation and semantic segmentation (§ 5.5) Finally, we find that our models produce higher-quality reconstructions using the pretraining decoder (§ 5.6).

## 5.1 MIMIC-3M OUTPERFORMS MULTIVIEW-HABITAT AND IMAGENET-1K ON DENSE GEOMETRIC TASKS

We finetune our trained models on two dense geometric tasks: NYUv2 depth estimation and Taskonomy surface normal prediction. We also finetune the CroCo models trained on MULTIVIEW-HABITAT using task-specific decoders adopted from MultiMAE and report their improved results.

Even though MIMIC-3M was generated automatically, without manual intervention, and uses no 3D annotations, representations pretrained on MIMIC-3M perform better on both dense geometric tasks (Table 1). These gains can be attributed to the inclusion of real sources–thanks to the flexibility of our method which allows us to use real-world videos of complex scenes as a data source.

We also validate the utility of multi-view correspondences by comparing MAE with CroCo models. CroCo offers significant gains over MAE on MIMIC-3M demonstrating the benefits of using correspondences during pretraining (Table 1) . In fact, CroCo when trained on MIMIC-3M leads to the state-of-the-art $\delta 1$ of 91.79 NYUv2 depth and L1 of 53.02 on surface normals using masked image modeling methods.

## 5.2 MIMIC-3M OUTPERFORMS THE MULTIVIEW-HABITAT AND REDUCES THE GAP TO IMAGENET-1K ON DENSE OBJECT TASKS.

To understand the potential of MIMIC for dense tasks which also require object-level understanding, we evaluate MAE and CroCo pretrained with MIMIC-3M on ADE20K semantic segmentation and MSCOCO pose estimation (Table 2). We observe consistent gains in comparison to the MULTIVIEW-HABITAT. We hypothesize that these improvements come from the real-world object-centric data from Objectron and Co3D. When compared to MULTIVIEW-HABITAT, MIMIC-3M reduces the performance gap by 7.36% with MAE and 2.64% with CroCo on manually curated, object-centric, and human-annotated ImageNet-1K.

## 5.3 SCALING UP MIMIC LEADS TO PERFORMANCE GAINS

We study the scaling trends of MIMIC by varying the data size. We experiment with two scales: the first MIMIC-1M with 1.3M image pairs and the second MIMIC-3M with 3.1M image pairs. We train CroCo with these two training sets and evaluate the performance on depth estimation (NYUv2), semantic segmentation (ADE20K), and surface normals (Taskonomy) (Table 3). We observe consistent improvements: $\delta 1$ by 2.33, mIOU on ADE20K by 3.73, and L1 loss by 4.10. We conjecture that the improvements occur because of the additional 1.8M image pairs added from three real datasets: CO3D, ArkitScenes, 3DStreetViews.

## 5.4 MIMIC REPRENSETATIONS IMPROVE WITH MORE PRETRAINING ITERATIONS

In contrast to models trained on MULTIVIEW-HABITAT, we do not observe performance saturation or degradation with pretraining iterations (see Figure 6 in their paper (Weinzaepfel et al., 2022)). Instead, the performance of both MIMIC-1M and MIMIC-3M improves on depth estimation and

Table 2: MIMIC-3M, our automatically generated dataset shows improvements over MULTIVIEW-HABITAT on dense object-related tasks such as ADE20K semantic segmentation and MSCOCO pose estimation. It even improves on ImageNet-1K classification and further closes the gap with models pre-trained on ImageNet-1K, curated with expensive crowdsourcing.

| Model | Pretraining dataset | ADE-20K(↑) | MSCOCO(↑) | | ImageNet-1K (↑) |
|-------|--------------------|-----------|-----------|-----|----------------|
| | | mIOU | AP | AR | % accuracy |
| MAE | MV-HABITAT | 40.30 | - | - | 32.50 |
| MAE | MIMIC-3M | **40.54** | 69.13 | 75.22 | **39.86** |
| CroCo | MV-HABITAT | 40.60 | 66.50 | 73.20 | 37.00 |
| CroCo | MIMIC-3M | **42.18** | **72.80** | **78.40** | **39.64** |
| | | +1.58 | +6.30 | +5.20 | +2.64 |
| MAE | ImageNet-1K | **46.10** | **74.90** | **80.40** | **67.45** |

Table 3: MIMIC-3M shows improvements over MIMIC-1M on depth estimation (NYUV2), Semantic Segmentation (ADE20K), Surface Normals Estimation (L1)

| Dataset | Frozen | NYUv2(↑) | ADE20K(↑) | Taskonomy(↓) |
|---------|--------|----------|-----------|--------------|
| | | $\delta 1$ | mIOU | L1 |
| MIMIC-1M | ✓ | 82.67 | 27.47 | 67.23 |
| MIMIC-3M | ✓ | **85.81** | **30.25** | **61.70** |
| | | +3.14 | +2.78 | -5.53 |
| MIMIC-1M | ✗ | 89.46 | 38.45 | 57.12 |
| MIMIC-3M | ✗ | **91.79** | **42.18** | **53.02** |
| | | +2.33 | +3.73 | -4.10 |

semantic segmentation (Figure 3(a)) for an iterations-matched training run. This trend holds regardless of whether the representations are fine-tuned or kept frozen.

## 5.5 MIMIC-3M OUTPERFORMS MULTIVIEW-HABITAT WITH FEW-SHOT FINETUNING

We measure the label efficiency of the learned representations trained on MIMIC-3M by evaluating its few-shot performance on NYUv2 depth estimation and ADE20K semantic segmentation. We freeze the image encoder and fine-tune the task-specific decoders by varying the number of training images. We run each k-shot finetuning at least 5 times and report the mean and the standard deviation of the runs. For depth estimation, we also experimented with k-shot regimes where k is less than 10. Overall the representations trained on our MIMIC-3M show better labeling efficiency than those trained using MULTIVIEW-HABITAT (Figure 3(b)). These gains can be attributed to the diverse, and real world training data during pretraining.

## 5.6 MIMIC ACHIEVES HIGHER FID SCORE AND LOWER RECONSTRUCTION ERROR

We analyze the quality of the reconstructions trained on MIMIC-3M versus MULTIVIEW-HABITAT. We use FID scores (Heusel et al., 2017), which indicate how realistic the reconstructions are and the reconstruction error (L2 loss) in the original masked image modeling objective. We sample a test set of 500 images from the Gibson dataset. We ensure that these images are sampled from the scenes that are exclusive of MULTIVIEW-HABITAT and MIMIC-3M pretraining datasets. We mask 90% of each test image and then compare the quality of the reconstructions (Table 4). Our analysis shows that CroCo trained on MIMIC-3M improves the FID by 12.65 points and reduces the reconstruction loss on the test set (see Appendix for visualizations).

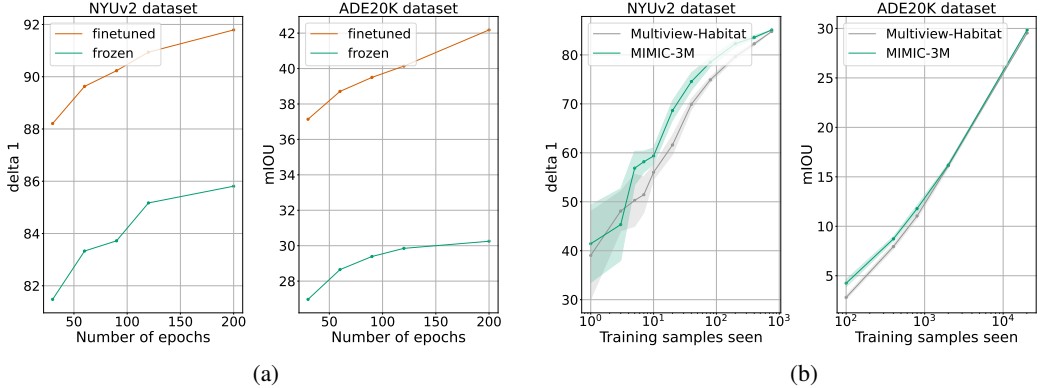

Figure 3: **(a)** CroCo pretrained on MIMIC shows an increasing trend with the number of training epochs. The figure on the left shows the trends for the fine-tuned and frozen versions of the encoder on NYUv2 depth estimation. The figure on the right shows the trend on the ADE20K dataset. **(b)** CroCo pretrained on MIMIC-3M achieves better few shot performance on CroCo pretrained on MULTIVIEW-HABITAT. The figure on the left shows the few shot performance on the NYUv2 dataset and the figure on the right shows the few shot performance on ADE20K.

Table 4: MIMIC-3M achieves better FID score and reduces the reconstruction loss on 500 test images from the Gibson dataset compared to MULTIVIEW-HABITAT

| Model | Dataset | Reconst. loss ($\downarrow$) | FID score ($\downarrow$) |
|-------|---------|------------------------------|--------------------------|
| CroCo | MV-HABITAT | 0.357 | 85.77 |
| CroCo | MIMIC-3M | **0.292** | **73.12** |

## 6 DISCUSSION

We present MIMIC, an approach to curate large-scale pretraining datasets from real-world videos and synthetic environments, geared towards dense vision tasks. Our work aims to provide a holistic solution that requires no manual intervention and domain knowledge about the data sources. We discuss below the limitations and safety considerations regarding our dataset and lay out opportunities for future work.

**Limitations.** There are several limitations of our work. First, we pretrain CroCo on MIMIC-3M using a fixed-sized architecture ViT-B/16; model scaling experiments are outside the scope of this work. Second, our curated dataset primarily consists of static objects and does not involve dynamic scenes. Lastly, MIMIC-3M has a small amount of object-centric data, and its suitability for object-related tasks is limited. Including more object-centric sources may help bridge this gap.

**Safety and ethical considerations.** While our method uses publicly available datasets for data curation, we acknowledge that the algorithm can be scaled up to scrape videos in the wild. We are aware of the privacy, and ethical issues caused by models trained on large-scale datasets and the amplification of the biases these models may result in. As such, we ensure to limit our data sources to only open-sourced video datasets. Lastly, we recommend the use of face blurring and NSFW filtering before scraping internet videos.

**Future work.** We would like to design methodologies to mine dynamic videos where epipolar geometric constraints do not apply, design new objectives for pretraining on image pairs curated using MIMIC, and evaluate representations on more diverse tasks. The flexibility of MIMIC makes it suitable for further scaling it up to even larger pretraining datasets.

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

## A  APPENDIX

## B  DATASET, RESOURCES, ASSETS

### B.1  DATASET USAGE

The code and instructions to download, access, and use MIMIC-3M can be found here. The primary use case of this dataset is to train a 3D-aware ViT in a self-supervised manner.

## B.2 COMPUTE RESOURCES

As mentioned in Section 4.1 (Pretraining) we train CroCo (Weinzaepfel et al., 2022) for 200 epochs, each epoch taking about 1 hour 40 minutes using 8 NVIDIA RTX A6000 GPUs. The cost for one training run is about 111 GPU days.

## B.3 ASSETS

We provide the details of the dataset and code licenses used in our study in Table 5. We bear all responsibility in case of violation of rights. Our code is primarily based on MAE (He et al., 2020), MultiMAE (Bachmann et al., 2022) and CroCo (Weinzaepfel et al., 2022) and our work is licensed under CC BY-NC-SA 4.0.

Table 5: List of the assets and licenses

| Asset | License |
|---|---|
| **Pretraining datasets** | |
| HM3D (Ramakrishnan et al., 2021a) | [link] |
| Gibson (Xia et al., 2018) | [link] |
| 3DStreetView (Zamir et al., 2016) | [link] |
| CO3D (Reizenstein et al., 2021) | [link] |
| Mannequin (Li et al., 2019) | [link] |
| ArkitScenes (Baruch et al., 2021) | [link] |
| Objectron (Ahmadyan et al., 2021) | [link] |
| ScanNet (Dai et al., 2017) | [link] |
| Matterport (Chang et al., 2017) | [link] |
| DeMoN (Ummenhofer et al., 2017) | [link] |
| **Downstream datasets** | |
| ImageNet-1K (Deng et al., 2009) | [link] |
| NYUv2 (Nathan Silberman & Fergus, 2012) | [link] |
| ADE20K (Zhou et al., 2019) | [link] |
| Taskonomy (Zamir et al., 2018) | [link] |
| MSCOCO (Lin et al., 2014) | [link] |
| **Code/Pretrained models** | |
| MAE (He et al., 2022) | [link] |
| CroCo (Weinzaepfel et al., 2022) | [link] |
| MultiMAE (Bachmann et al., 2022) | [link] |

## C DATA CURATION DETAILS

### C.1 DETAILS ON MINING POTENTIAL PAIRS

We utilized different data types within our datasets, including videos, 3D scenes, and street views. Consequently, the process of mining potential pairs for each data type varied. For street views (Zamir et al., 2016), we adopted a strategy where we grouped images based on their target id (images that have the same target id in their name, show the same physical point in their center). Subsequently, among all possible combinations of images in a group, we selected the pair with minimal overlap ranging from 50% to 70%.

When dealing with video data, a practical approach involved creating a list of frames at regular time intervals, determined by the speed of the video. Then, we generated pairs of consecutive frames from this list. In cases where substantial overlap between consecutive frames was observed, we specifically chose the second consecutive frame and evaluated its overlap with the preceding frame. We implemented this step to ensure that the selected frame pair exhibits an appropriate level of dissimilarity and minimized redundancy.

To tackle the challenges associated with handling 3D scenes, we employed the habitat simulator (Savva et al., 2019) to sample locations within the navigable area of the scene. We initialized an agent with a random sensor height and rotated it eight times at $45°$ intervals, capturing a comprehensive view of the surroundings to form the first list of eight images. Subsequently, we sampled

a random rotation degree from multiples of $60°$ (excluding $180°$ and $360°$), and rotated the agent accordingly before moving in the current direction for a random step ranging from 0.5 to 1 meter. We repeated the process of rotating eight times at $45°$ intervals, capturing the second list of eight images. Likewise, we randomly rotated and moved the agent to generate the third list of eight images. From these lists, we selected an optimal pair $(img_1, img_2)$ from a pool of $8 \times 16$ potential pairs. $img_1$ belonged to the first list, while $img_2$ was chosen from the combined pool of the second and third lists, with a minimal overlap ranging from 50% to 70%, if applicable.

The selection of a $45°$ rotation aimed to capture a comprehensive view of the environment while minimizing redundancy. Furthermore, the choice of rotation degrees as multiples of $60°$ prevented capturing images in directions already covered by those obtained with the $45°$ rotation, effectively avoiding the capture of zoomed-in versions of previously acquired images.

## C.2 DETAILS ON MEASURING THE OVERLAP

Given a pair of images or views from a scene (we call it a potential pair), we checked whether these two are sufficiently overlapped during the six steps. If they had enough overlap, we saved this pair along with other metadata for the next phase, which was the model pretraining. The six steps are listed below:

**Keypoint localization using SIFT (Lowe, 2004).** We used SIFT (Scale-Invariant Feature Transform) as a feature detector to localize the two views' key points separately. SIFT has been shown to perform well compared to other traditional methods. Figure 4a provides an example pair with key points.

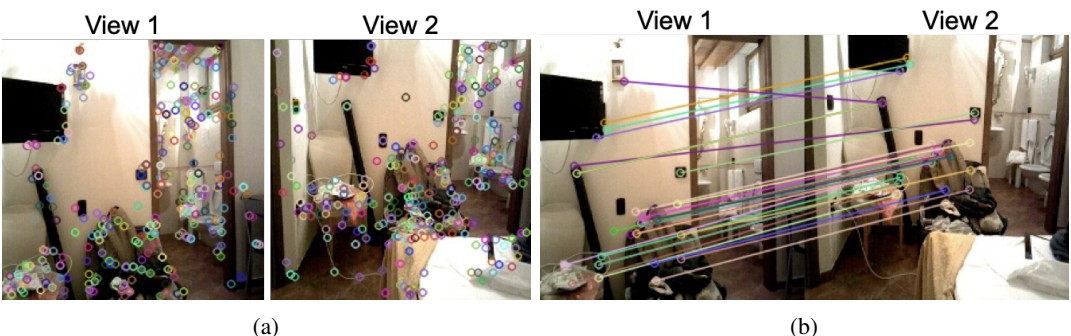

Figure 4: **(a)** A pair of images with SIFT key points. **(b)** Matching key points of images with a brute force matcher.

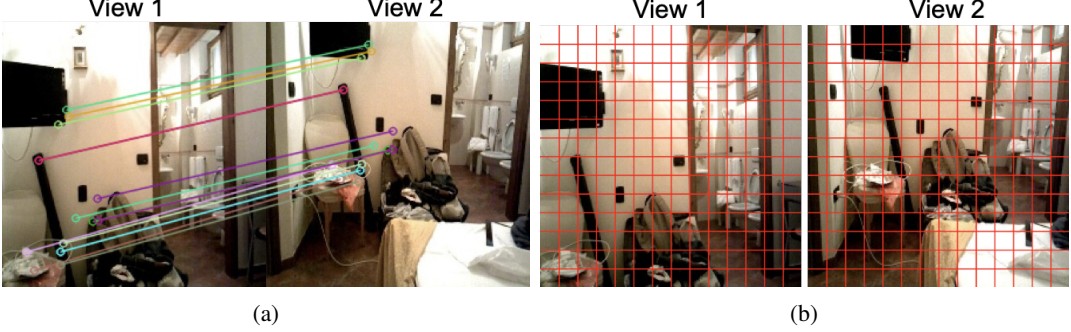

Figure 5: **(a)** Inlier matches after finding the homography matrix. **(b)** Dividing each image to non-overlapping patches.

**Brute force matching.** Having obtained both key point features and their descriptors from the previous step, we performed a brute-force matching process to match the key points in the first view (source points) with the key points in the second view (destination points). We present matches between two views in Figure 4b.

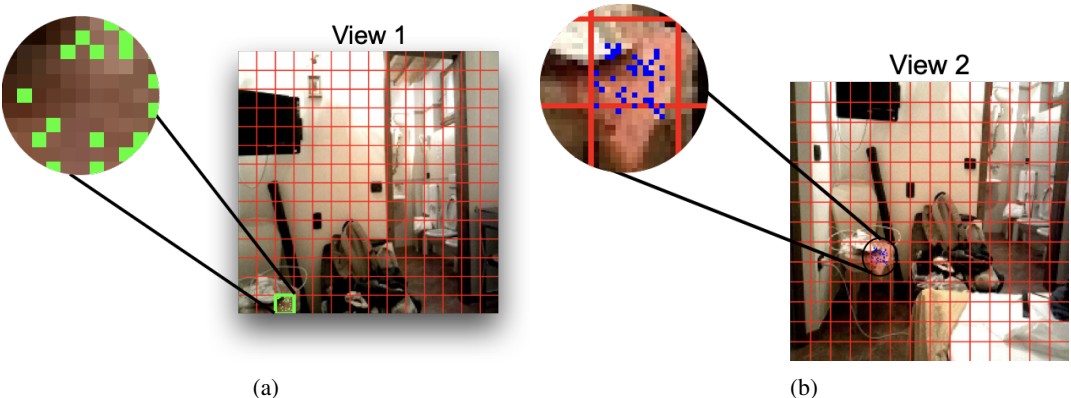

Figure 6: **(a)** Sampling random points from a patch in the first view. **(b)** Blue points are the corresponding points of the green points in the second view.

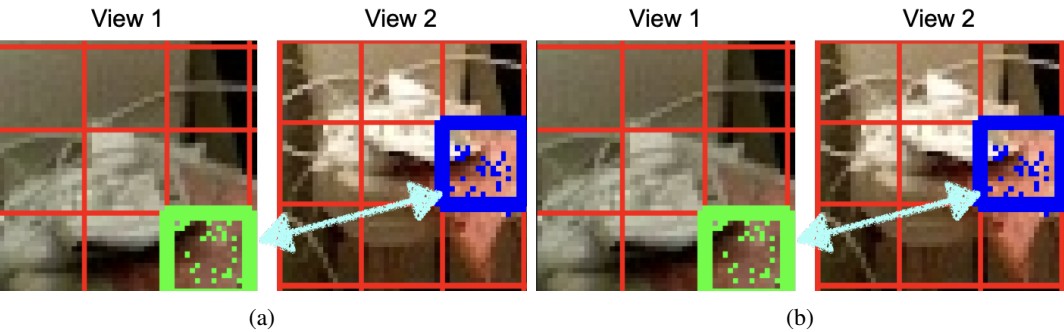

Figure 7: **(a)** The green patch from the view 1 is matched with the blue patch in view 2. **(b)** Two views with their matching patches (matching patches have the same color).

**Finding homography transformation (Hartley & Zisserman, 2003).** We leveraged the homography (Hartley & Zisserman, 2003) matrix to translate the transformation among the views with provided source and destination points matches from the previous step. However, we know the found transformation is not thoroughly accurate and free of errors. Therefore, to overcome this issue, we used RANSAC (Fischler & Bolles, 1981) to conclude with better estimations of the transformation. As a result, only some of the matches was categorized as inliers. Inlier matches are shown in Figure 5a

**Creating non-overlapping patches.** After finding the homography matrix, we divided each view into non-overlapping patches ($16 \times 16$ here) and matched patches from view 1 to view 2, see Figure 5b.

**Obtaining the patch correpondences** To find a corresponding patch in the second view for a particular patch in the first view, we performed the following steps: 1. Randomly sampled a suitable number of points within the specific patch in the first view (e.g., 100 points). In Figure 6a, random green points are sampled within the green patch of the first view. 2. Applied the homography matrix $H$ to the sampled points to determine their corresponding positions in the second view. 3. Determined the patch number in which each corresponding point falls, such as $patch(x = 17, y = 0) = 1$. 4. Identified the patch that contains the maximum number of corresponding points as the match for the specific patch in the first image. In Figure 6b, the blue points represent the positions of the corresponding points in the second view that fall within nearby patches. It can be observed that the majority of the blue points cluster within a specific patch, which is marked as the matched patch for the green patch. This match is illustrated in Figure 7a.

**Measuring the visual overlap** We repeated the procedure from the previous step for all patches in the first view to determine their matches in the second view. We computed the count of patches in the first view that have a matching patch within the boundaries of the second view, provided that the matching patch has not been previously matched with another patch from the first view. Then, we divided this count by the total number of patches, serving as a metric to measure the overlap.

To ensure a comprehensive evaluation, we performed the mentioned algorithm both for finding $overlap(view1, view2)$ and its inverse, $overlap(view2, view1)$. We chose the minimum value between these two overlap metrics as the final overlap measure.

Subsequently, we retained pairs with an overlap ranging from 50% to 75% along with corresponding patches information. Figure 7b showcases all patches from the first view that have their matches falling within the second view. Additionally, Figure 8 provides an illustrative example of a retained pair of images from each dataset, along with their corresponding patches.

# D  DOWNSTREAM TASKS

## D.1  FINETUNING DETAILS

For fine-tuning depth estimation, semantic segmentation, and surface normal estimation we adopt the task-specific decoders from MultiMAE Bachmann et al. (2022). For pose estimation, we use the ViTPose Xu et al. (2022) decoders. In Table 6 , we provide the details of the hyperparameters used for finetuning CroCo (Weinzaepfel et al., 2022) pretrained on MIMIC-3M on NYUv2 (Nathan Silberman & Fergus, 2012), ADE20K (Zhou et al., 2019), Taskonomy (Zamir et al., 2018), MSCOCO (Lin et al., 2014).

## D.2  ERROR ESTIMATES

To estimate the variability associated with our fine-tuned models we compute the error estimates for each of our fine-tuned models. Specifically, we create 100 test sets from each of the downstream (val/test) datasets by sampling with replacement and then report the minimum, maximum, mean, and standard deviation of the metric in Table 7. Overall we observe that the mean values are close to the numbers reported in the main paper and the standard deviation is small.

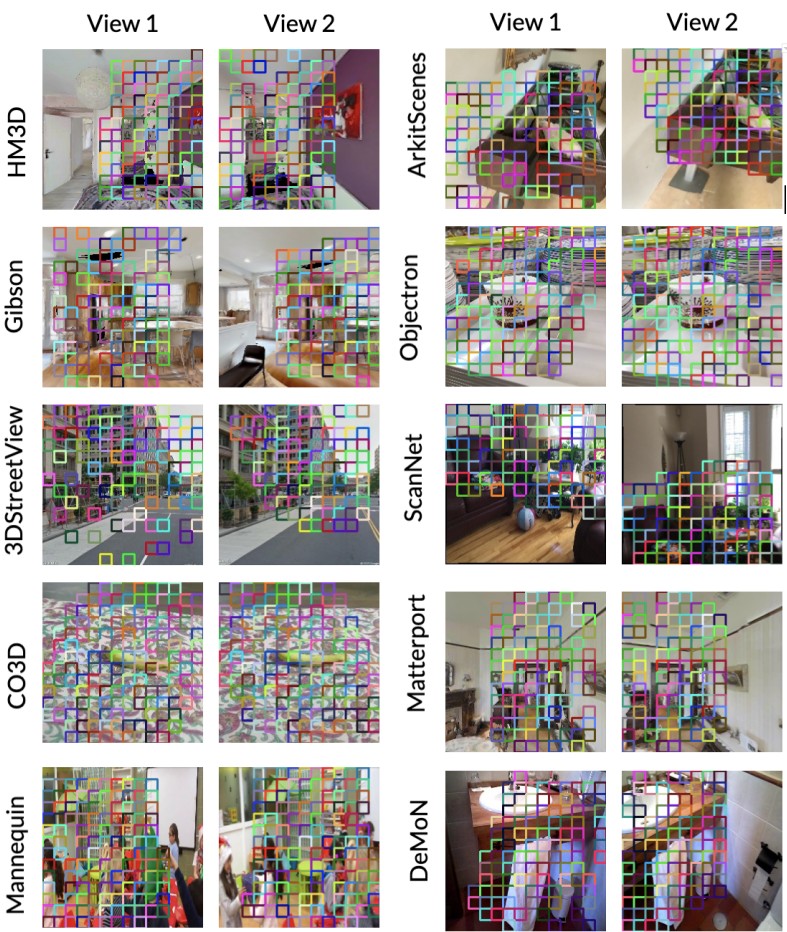

Figure 8: Visualizations of the patchwise correspondences (matching patches have the same color).

Table 6: Hyperparameters used for fine-tuning NYUv2 (depth estimation), ADE20K (semantic segmentation), Taskonomy (surface normals)

| Hyperparameter | NYUv2(depth) | ADE20K(sem.seg.) | Taxonomy (surf.norm.) | MSCOCO(pos.est.) |
|---|---|---|---|---|
| Optimizer | AdamW | AdamW | AdamW | AdamW |
| Learning rate | 0.0001 | 0.0005 | 0.0003 | 0.0005 |
| Layer-wise lr decay | 0.75 | 0.75 | 0.75 | 0.75 |
| Weight decay | 0.0003 | 0.05 | 0.05 | 0.1 |
| Adam $\beta$ | (0.9, 0.999) | (0.9, 0.999) | (0.9, 0.999) | (0.9, 0.999) |
| Batch size | 64 | 16 | 8 | 512 |
| Learning rate schedule. | Cosine decay | Cosine decay | Cosine decay | Linear Decay |
| Training epochs | 2000 | 64 | 100 | 210 |
| Warmup learning rate | - | 0.000001 | 0.000001 | 0.001 |
| Warmup epochs | 100 | 1 | 5 | 500 |
| Input resolution | $256 \times 256$ | $512 \times 512$ | $384 \times 384$ | $224 \times 224$ |
| Augmentation | ColorJitter, RandomCrop | HorizontalFlip, ColorJitter | - | TopDownAffine |
| Drop path | 0.0 | 0.1 | 0.1 | 0.30 |

## D.3 VISUALIZATIONS OF THE FINE-TUNED MODELS

In this section, we provide the visualizations of the depth maps, semantic segmentation masks, surface normal predictions, and pose regression outputs after finetuning CroCo pretrained using MIMIC-3M. For finetuning NYUv2 for depth, ADE20K for semantic segmentation, and Taskonomy for surface normals, we followed MultiMAE (Bachmann et al., 2022) and used the settings from D.1. For finetuning on MS COCO we used ViTPose (Xu et al., 2022).

Table 7: Error estimates for fine-tuning NYUv2 depth, ADE20K semantic segmentation, Taskonomy surface normal prediction

| Task(metric) | Dataset (Val/Test) | Min | Max | Standard Deviation | Mean | Reported value |
|---|---|---|---|---|---|---|
| Depth Estimation ($\delta_1$) | NYUv2 (Silberman et al., 2012) | 90.17 | 92.91 | 0.56 | 91.70 | 91.79 |
| Semantic Segmentation (mIOU) | ADE20K (Zhou et al., 2019) | 39.75 | 43.36 | 0.75 | 41.71 | 42.18 |
| Surface Normal Estimation (L1) | Taxonomy (Zamir et al., 2018) | 48.28 | 54.09 | 1.24 | 50.78 | 53.02 |

**Depth Estimation.** Figure 9 shows the input RGB file, predicted depth maps, and ground truth depth maps from the validation set after finetuning on NYUv2.

**Semantic Segmentation.** Figure 10 shows the RGB images, predicted semantic segmentations, and the ground truth labels from the ADE20K validation set after finetuning.

**Surface Normals.** Figure 11 shows predicted surface normals from the Taskonomy test set after finetuning.

**Pose estimation.** Figure 12 shows the predicted keypoints from MS COCO validation set after finetuning.

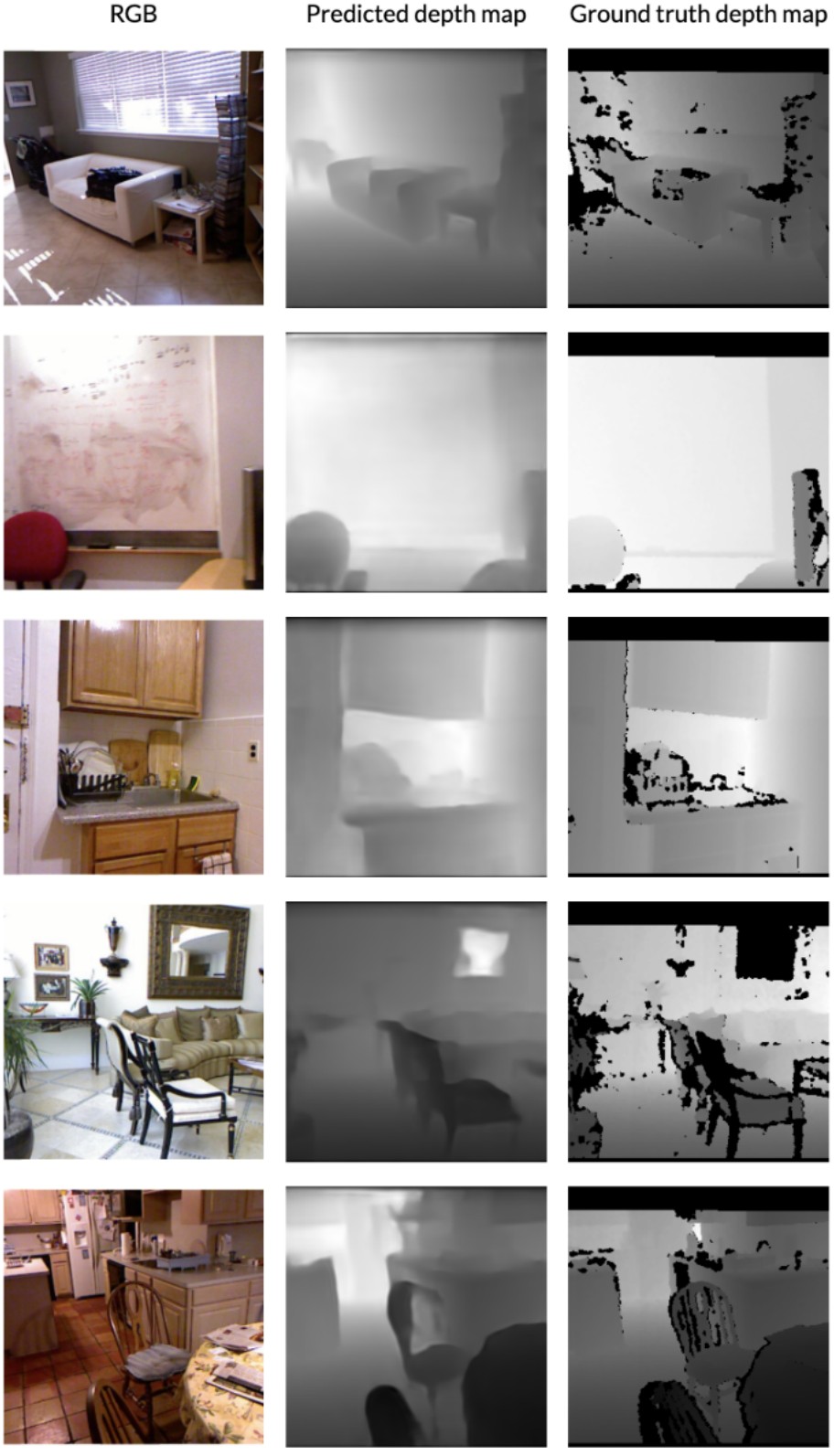

Figure 9: Visualizations of the depth maps

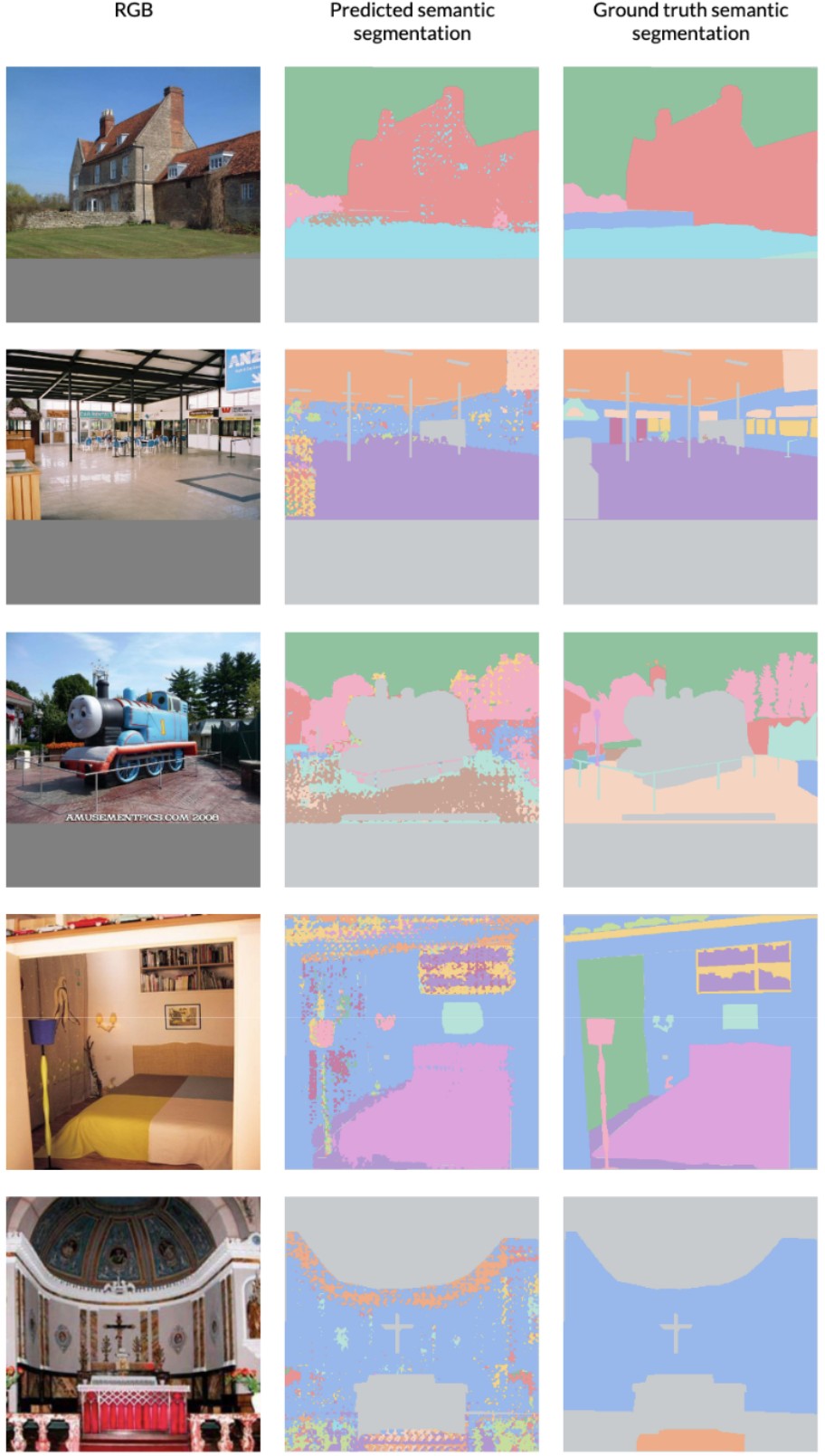

Figure 10: Visualizations of the segmentation maps

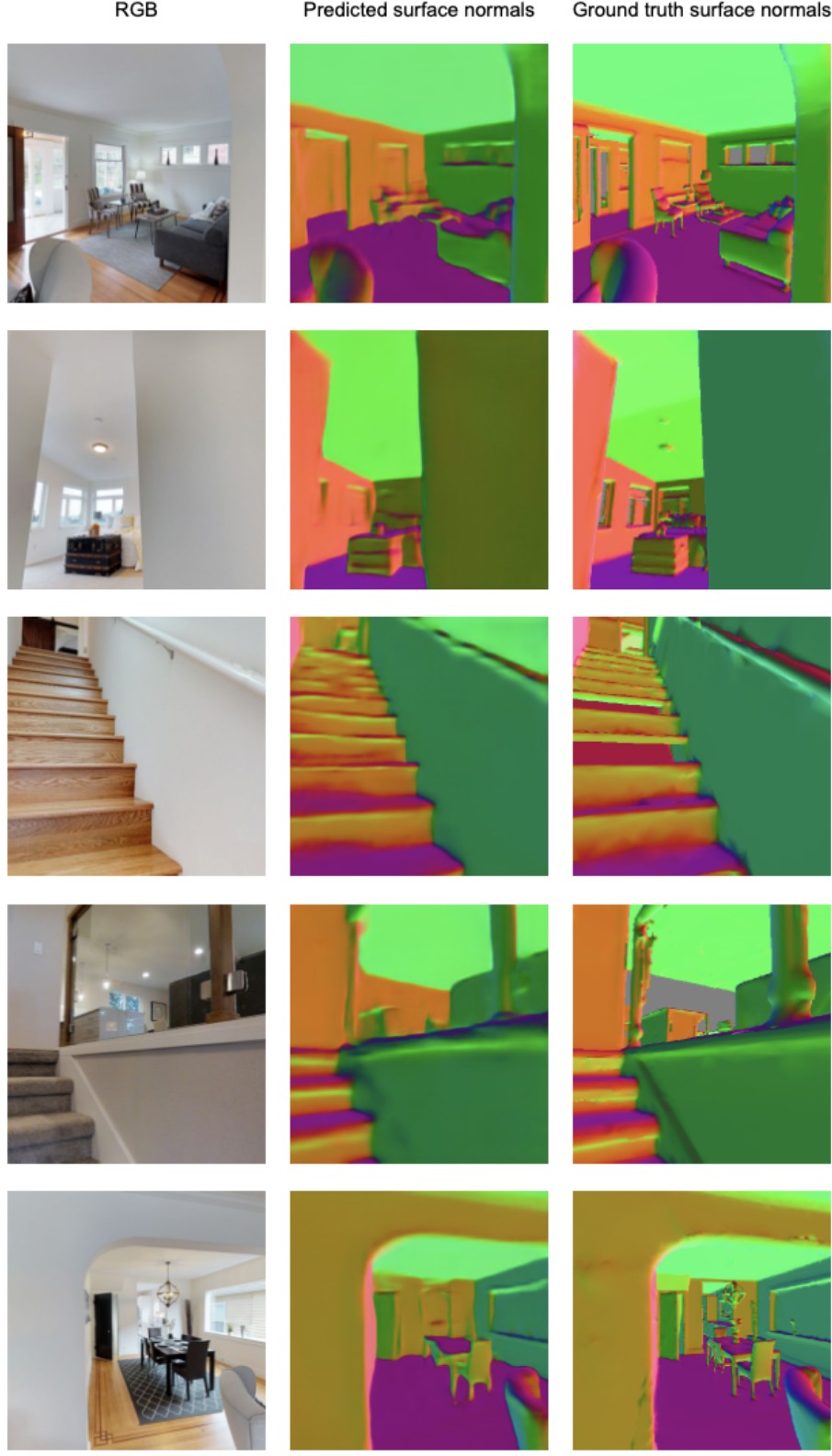

Figure 11: Visualizations of the surface normal predictions

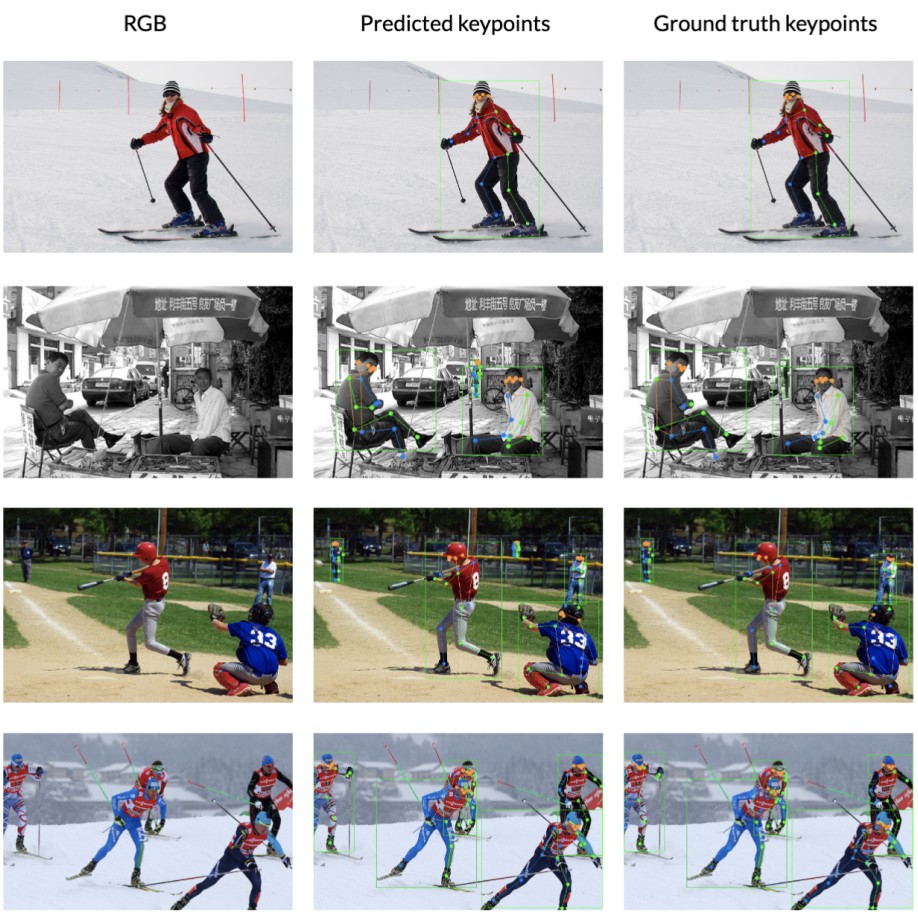

Figure 12: Visualizations of the pose estimation

