# OpenReview forum: "MIMIC: Masked Image Modeling with Image Correspondences"
_ICLR.cc/2024/Conference — ICLR 2024 Conference Withdrawn Submission_

### Official Review · Reviewer_ytkL · 2023-10-28

**Soundness:** 2 fair
**Presentation:** 3 good
**Contribution:** 2 fair
**Rating:** 3
**Confidence:** 4

**Summary:**

This submission studies self-supervised image representation learning. It follows CroCo, a method that extends MAE to cross-view completion. The methods benchmarked are still MAE and CroCo, so the focus is not algorithmic contribution. The argument is that CroCo builds the dataset using ground truth camera pose while this method does not. Looking into details, the method uses image correspondence to filter pairs that have the about-right overlap for training. This leads to a dataset with 70% synthetic data from Habitat and 30% real dataset from various datasets. The synthetic part is not largely different from CroCo dataset (or say MV-Habitat) except for the pair is selected according to correspondence-based pose instead of ground truth pose. The real data part is good but occupies a limited portion in the whole dataset. Bencharmking on NYU depth and Taskonomy normal shows some gains. Many related evaluations are not covered.

**Strengths:**

+ The authors visit several datasets to construct the real part. This is an effort I appreciate.

**Weaknesses:**

- The evaluation part lacks empirical sginificance. I think the most interesting thing about CroCo is to train useful representations for geometric tasks. Only experiments on NYU depth and Taskonomy subset normal are conducted. More geometric tasks are suggested, including single view pose regression, two-view correspondence, other geometric understanding tasks on Taskonomy like occlusion edge, single-view reconstruction for objects and scenes.
- Significance on non-geometric tasks like ADE and COCO is not convincing. Please directly compare with EVA or InternImage.
- Technical contribution is limited. The method is still CroCo. Instead of representation from image correspondence, I would like to say it's *** representation from cross-view completion on pairs selected by correspondence***. The title is over-claiming.

Sorry I cannot vote acceptance due to limited technical or empirical contributions.

**Questions:**

See the weakness box.

---

### Official Review · Reviewer_fi3r · 2023-11-01

**Soundness:** 3 good
**Presentation:** 3 good
**Contribution:** 2 fair
**Rating:** 3
**Confidence:** 4

**Summary:**

The main contribution of the paper is to construct a large-scale multi-view dataset MIMIC-3M with 3.1M multi-view image pairs for representation learning. The paper pre-trains MAE and CroCo on MIMIC-3M and tests on multiple benchmark datasets. The paper show improvements on NYU-Depth v2 and Taskonomy surface normal compared to models pre-trained on ImageNet 1K.

**Strengths:**

1. The paper proposes a way to select image pairs from existing datasets that could potentially train better models for dense prediction tasks.

2. The paper is in general easy to read. The paper contains details on the implementation.

3. The paper shows promising results on multiple benchmarks such as NYU-v2 and Taskonomy surface normal.

**Weaknesses:**

1. For depth estimation, the model is only tested on NYU-v2, which is also a dataset containing mostly indoor scenes. So I feel that the current experiments are not convincing enough to support the claim that pre-training on the proposed dataset is better for depth estimation in-the-wild. How about testing on datasets that contain more general images, such as KITTI, TUM RGBD, Sintel?

2. Since the paper claims that pre-training on the constructed MIMIC-3M dataset is better for dense prediction, I feel that testing on optical flow on multiple datasets is also necessary to support the claim.

3. In Table 2, the model trained on the proposed dataset does not outperform the ImageNet 1K benchmark on all tasks. This makes it less convincing to train the proposed dataset. One way to make the paper stronger would be to increase the scale of the dataset, maybe 5M or 10M? Would that outperform ImageNet 1K? Since the proposed way of constructing pairs of images is labor-free compared to constructing ImageNet, maybe this would make the argument of using the proposed dataset stronger.

4. Is it possible to crawl online videos? Maybe this could increase the diversity of the proposed dataset a lot.

5. There are very few image pairs visualizations in the paper. Since the dataset is the main contribution, it might be helpful to include more visual results.

6. Why is the overlapping ratio determined to be 50%-70%, is there analysis or ablation studies?

7. Will the dataset be released?

**Questions:**

Please see my questions above. My main concern is that the paper may need more experiments to support the claim.

---

### Official Review · Reviewer_AdG9 · 2023-11-02

**Soundness:** 4 excellent
**Presentation:** 3 good
**Contribution:** 4 excellent
**Rating:** 8
**Confidence:** 4

**Summary:**

The paper proposes a method for generating large scale paired image datasets from real or synthetic video data. The method uses SIFT matching to identify spatially related patches between video frames, which then become pairs that can be used for representation learning via MAE, CroCo, etc. They build a dataset, pre-train a representation, then evaluate the representation on downstream tasks.

**Strengths:**

- Being able to mine large data for pairs is a relevant task in representation learning.

- They outperform multiview habitat on their evaluations.

- The method, being based on classical techniques like SIFT and RANSAC, should scale well.

- The method is simple but effective.

**Weaknesses:**

- (minor) some exposition on what the

- This paper might be better suited for a computer vision venue

- The paper targets dense vision tasks, but it would be interesting to see the method used to generate pairs for constrastive learning, as well as evaluations on non-dense tasks such as imagenet finetuning/linear probe accuracy.

**Questions:**

Is there any reason this wouldn't be useful for general representation learning beyond dense tasks?

---

### Official Review · Reviewer_HFtz · 2023-11-03

**Soundness:** 3 good
**Presentation:** 3 good
**Contribution:** 3 good
**Rating:** 6
**Confidence:** 4

**Summary:**

This paper introduces a curation approach for multi-view datasets, dubbed MIMIC, from videos and 3D simulated environments. The proposed approach samples potential image pairs from data sources, measures the overlap of each pair via classical matching algorithms, and filters out the degenerated matches. Finally, two scales of dataset, MIMIC-1M and MIMIC-3M, are constructed. Experiments results with MAE and CroCo models show that pre-training with the proposed MIMIC datasets shows better performance on dense geometric tasks than either ImageNet-1K or multi-view Habitat. The fine-tuning experiments show that pre-training with the proposed MIMIC dataset helps object-related tasks. However, due to a lack of object-centric properties in MIMIC, MAE pre-trained with ImageNet-1K still outperforms the one pre-trained with MIMIC.

**Strengths:**

[Writing quality] The paper is well-written and easy to follow. Figure 1 illustrates the overview of the data curation process used to build MIMIC in detail. The authors also provided implementation details including hyper-parameters used for experiments.

[Soundness of the method] The proposed curation approach seems reasonable. As the classical matching algorithm using SIFT features does not require training, it is generalizable and suitable for matching data from multiple data sources. The effectiveness of these matching schemes is also illustrated in Figures 4, 5, 6, and 7 of the appendix.

[Experiment results] The proposed multi-view datasets, MIMIC, significantly improve the performance of models on both the dense geometric and dense object-related tasks, especially when it comes to cross-view masked image modeling (CroCo).

**Weaknesses:**

[Overlap measurement] The authors used a patch size 16, consistent with the size used for ViT, to determine the overlap between two images. It raises the question of whether a mismatch between the patch size used for overlap measurement and the one used for masked image pre-training could present challenges. For instance, is it feasible to conduct pre-training on a model using a patch size of 32 on MIMIC (while the image pairs are computed using a patch size 16)? How much the performance would degrade due to the mismatch?

[Minor] There are several misusages of \cite, where you should use \citep, on pages 2 and 3. Please check the usage of \citep to improve the readability.

[Minor] There are some outdated references; BeiT was published at ICLR 2022, not the arXiv 2021; CroCo was published at NeurIPS 2022, not the arXiv 2022; and ViT was published at ICLR 2021 not the arXiv 2020.

**Questions:**

Please refer to the weakness section.

---

### Author Response · Authors · 2023-11-21
**Withdrawing MIMIC**

We are grateful to the reviewers for taking the time to review our work and for their valuable suggestions. We acknowledge the reviewers' concerns and would like to withdraw our paper.

Thanks,

MIMIC team